# Health-Related Quality of Life following Cytoreductive Radical Prostatectomy in Patients with De-Novo Oligometastatic Prostate Cancer

**DOI:** 10.3390/cancers13225636

**Published:** 2021-11-11

**Authors:** Michael Chaloupka, Lina Stoermer, Maria Apfelbeck, Alexander Buchner, Vera Wenter, Christian G. Stief, Thilo Westhofen, Alexander Kretschmer

**Affiliations:** 1Department of Urology, LMU Klinikum, Ludwig-Maximilians University Munich, 81377 Munich, Germany; michael.chaloupka@med.uni-muenchen.de (M.C.); L.Stoermer@campus.lmu.de (L.S.); maria.apfelbeck@med.uni-muenchen.de (M.A.); alexander.buchner@med.uni-muenchen.de (A.B.); christian.stief@med.uni-muenchen.de (C.G.S.); thilo.westhofen@med.uni-muenchen.de (T.W.); 2Department of Nuclear Medicine, LMU Klinikum, Ludwig-Maximilians University Munich, 81377 Munich, Germany; vera.wenter@med.uni-muenchen.de

**Keywords:** cytoreductive, HRQOL, metastatic prostate cancer

## Abstract

**Simple Summary:**

Recent retrospective data indicate a survival benefit of surgical removal of the prostate in patients with oligometastatic prostate cancer as part of a multimodal therapeutic regime. However, the impact of radical prostatectomy (RP) on patient’s health-related quality of life (HRQOL) in this scenario has not been evaluated yet. In a contemporary and well-balanced cohort, we compared the self-assessed HRQOL of patients with oligometastatic prostate cancer with patients with locally confined prostate cancer two years postoperatively. In multivariate analysis, we found no significant difference in postoperative HRQOL and urinary continence recovery in patients with de-novo oligo-metastatic compared to patients with locally confined prostate cancer.

**Abstract:**

(1) Background: local treatment of the primary tumor has become a valid therapeutic option in de-novo oligo-metastatic prostate cancer (PC). However, evidence regarding radical prostatectomy (RP) in this setting is still subpar, and the effect of cytoreductive RP on postoperative health-related quality of life (HRQOL) is still unclear. (2) Methods: for the current study, patients with de-novo oligo-metastatic PC (cM1-oligo), defined as ≤5 bone lesions in the preoperative staging, were included, and matched cohorts using the variables age, body-mass index (BMI), and pT-stage were generated. Patient-reported outcome measures (PROMS) were assessed pre- and postoperatively using the validated EORTC-QLQ-C30, IIEF-5, and ICIQ-SF questionnaires. The primary endpoint for univariate and multivariable analysis was good general HRQOL defined by previously validated cut-off values. (3) Results: in total, 1268 patients (*n* = 84 (7%) cM1-oligo) underwent RP between 2012 and 2020 at one tertiary care center. A matched cohort of 411 patients (*n* = 79 with oligo-metastatic bone disease (cM1-oligo) and *n* = 332 patients without clinical indication of metastatic disease (cM0)) was created. The median follow-up was 25mo. There was no significant difference in good general HRQOL rates between cM1-oligo-patients and cM0-patients before RP (45.6% vs. 55.2%, *p* = 0.186), and at time of follow-up (44% vs. 56%, *p* = 0.811). Global health status (GHS) worsened significantly in cM0-patients compared to baseline (−5, *p* = 0.001), whereas GHS did not change significantly in cM1-oligo-patients (+3.2, *p* = 0.381). In multivariate analysis stratified for good erectile function (IIEF5 > 18; OR 5.722, 95% CI 1.89–17.36, *p* = 0.002) and continence recovery (OR 1.671, 95% CI 1.03–2.70, *p* = 0.036), cM1-oligo was not an independent predictive feature for general HRQOL (OR 0.821, 95% CI 0.44–1.53, *p* = 0.536). (4) Conclusions: in this large contemporary retrospective analysis, we observed no significant difference in HRQOL in patients with the oligometastatic bone disease after cytoreductive radical prostatectomy, when compared to patients with localized disease at time of surgery.

## 1. Introduction

In recent years, a plethora of systemic therapeutic options for patients with advanced prostate cancer have emerged [1,2,3,4,5,6]. Furthermore, large randomized trials showed a survival benefit of radiation therapy of the primary tumor in patients with low-volume metastatic disease [7,8,9]. These data led to the recommendation of current guidelines to offer androgen deprivation therapy (ADT) combined with radiotherapy of the prostate to patients presenting with low-volume metastatic prostate cancer as part of a multimodal therapy regime [10]. In contrast, evidence for surgical removal of the primary tumor (radical prostatectomy, RP) is sparse, and consequently, RP is not recommended by current guidelines outside of clinical trials. It has been described that RP significantly impacts postoperative health-related quality of life (HRQOL) in the respective patients. Hereby, studies assessing self-reported HRQOL showed that patients who underwent RP had a higher prevalence of urinary incontinence and erectile dysfunction compared to radiotherapy and observation, respectively [11]. In addition, it has been shown that the risk of subpar functional outcomes correlates with the local tumor stage [12]. To date, there is no analysis focused on HRQOL after cytoreductive RP. Considering the fact that many patients with de-novo metastatic bone-disease also present with locally advanced cancer but are otherwise asymptomatic, this raises the question of whether offering cytoreductive RP might cause unnecessary harm to the respective patients.

To address this paucity of data, we performed a comprehensive analysis of the effect of cytoreductive RP on the HRQOL in a contemporary and well-balanced cohort of patients with oligometastatic prostate cancer.

## 2. Materials and Methods

### 2.1. Patient Population, Study Design, and Data Assessment

Inclusion criteria for the current study encompassed: biopsy-proven prostate cancer, history of RP at one tertiary center and completed follow-up. Patients with preoperative ADT and pre-RP radiotherapy of the prostate were excluded from further analysis. Between July 2012 and September 2020, 1268 patients (*n* = 800 open retropubic, *n* = 468 robot-assisted laparoscopic) fulfilled these criteria and were considered for analysis. Surgical standard procedures at our department have been described before [13]. A total of 84 of 1268 (7%) patients had suspicion of oligometastatic bone disease (cM1-oligo) based on preoperative imaging at the time of RP. CM1-oligo was defined as ≤5 bone lesions in the preoperative staging. Consequently, patients with nodal-only metastatic disease (cM1a), as well as visceral metastases (cM1c), did not undergo further analysis. All patients were informed about and consented in the experimental and individual multimodal therapeutic approach.

After approval by the local ethics committee (#20-1022), PROMs were prospectively documented pre- and postoperatively. Hereby, questionnaires were sent by mail to eligible patients. Erectile function was assessed using the validated International Index of Erectile Function (IIEF-5), defining good erectile function with IIEF-5 score ≥ 18 [14]. Urinary continence was assessed using the validated short form of the International Consultation on Incontinence Questionnaire (ICIQ-SF). With scores ranging from 0 to 21, higher scores indicate worsening of urinary continence [15]. Continence recovery was defined as the usage of up to one security pad per day. HRQOL was assessed using the validated European Organization for Research and Treatment of Cancer (EORTC) quality of life questionnaire (QLQ)-C30. According to EORTC guidelines, general HRQOL was assessed using the global health status (GHS) domain of the QLQ-C30 questionnaire. Good general HRQOL was defined as a GHS ≥70 following previously published cut-off values [16]. For QLQ-C30 functioning scores, higher scores indicate better functioning. For QLQ-C30 symptom scores, higher scores indicate a greater impact on the respective symptom. Perioperative patient characteristics were assessed by analysis of the respective institutional medical records.

### 2.2. Statistical Analysis

Based on the clinical variables “age”, "BMI," and “pT stage”, 2 matched cohorts were generated. The 3 variables were chosen based on previously published studies indicating that these variables represent independent risk factors for postoperative impairment of HRQOL and functional outcome [12,17,18]. Due to the limited sample size of cM1-oligo patients, the matching process was limited to 3 variables and not expanded throughout all variables that showed significant differences in the unmatched cohort. Consequently, a matched cohort of 411 patients (*n* = 79 with oligometastatic bone disease and *n* = 332 without signs of metastatic disease) was created and considered for further analysis. 

Testing of normal distribution was performed by the Shapiro–Wilk test. Statistical analysis was performed using SPSS V28 software (IBM, Armonk, NY, USA). Continuous variables were presented as means and standard deviation as well as median with the interquartile range as indicated. Categorical variables were presented as absolute numbers and percentages. Fisher’s exact test and Chi-square test were used for univariate analysis of categorical data. The Mann–Whitney U test was used for univariate analysis of continuous data. The primary endpoint for univariate and multivariable analysis was good general HRQOL with a GHS ≥ 70. For multivariable analysis, Cox regression models and binary logistic regression models were used. For univariate survival analyses, Kaplan–Meier curves were generated, and log-rank testing was performed. A *p*-value of <0.05 was considered statistically significant.

## 3. Results

### 3.1. Patient Characteristics and Survival Outcomes

Detailed patient characteristics of the unmatched and matched cohorts are displayed in Table 1.

Briefly, preoperative tumor burden was generally higher in the unmatched cM1-oligo cohort compared to unmatched cM0-patients with significantly higher median PSA levels, Gleason grade group 8–10, and positive lymph node disease (*p* < 0.001, respectively).

In the next step, a matched cohort was created as described above. The median follow-up for the entire cohort was 25 months.

Regarding the cM1-oligo cohort, 28 patients (35.4%) were staged using PSMA-PET/Ct, whereas staging was based on conventional imaging in 51 patients (64.6%). Regarding the cM0 cohort, 27 patients (9.7%) were staged using PSMA-PET/Ct, whereas staging was based on conventional imaging in 192 patients (68.8%). A total of 53 patients (16%) of the cM0-cohort presented a low-risk profile and did not undergo preoperative staging.

Notably, the ratio of robot-assisted laparoscopic radical prostatectomies was significantly lower in cM1-oligo patients compared to cM0-patients (12.7% vs. 34.9%, *p* < 0.001). Furthermore, patients in the cM1-oligo subgroup showed significantly higher rates of history of TUR-P (12.7 vs. 3.0%, *p* = 0.001) and a significantly higher rate of patients with an American Society of Anesthesiology (ASA) score of ≥3 (60.9 vs. 44.6%, *p* = 0.017). There were no differences in rates of history of hernia surgery between both groups (*p* = 0.723) and the median number of lymph nodes removed during the surgical procedure (*p* = 0.124).

Regarding adjuvant and salvage therapies, radiotherapy rates (22.6% vs. 24.1%, *p* = 0.401) after surgery did not vary significantly between cM1-oligo and cM0-patients. However, rates of perioperative androgen deprivation therapy were significantly higher for cM1-oligo patients (53.8 vs. 35.4%, *p* = 0.016), as well as rates novel anti-androgen therapies (10.1 vs. 2.7%, *p* = 0.003), and taxane-based chemotherapy (6.3 vs. 1.0%, *p* < 0.001). Survival analysis revealed a significantly shorter estimated 5-year cancer-specific survival rate (61% vs. 81%, *p* = 0.006) and a shorter estimated 5-year overall-survival rate (38% vs. 57%, *p* = 0.004) for cM1-oligo patients compared to cM0-patients (Appendix A).

### 3.2. Functional Outcomes

Pre- and postoperatively assessed functional outcomes are displayed in Table 2. In summary, we observed significantly lower total IIEF-5 scores in cM1-oligo patients compared to cM0-patients preoperatively as well as at the time of follow-up (8.5 (SD 10.2) vs. 11.3 (9.9), *p* = 0.022; 1.3 (4.2) vs. 3.5 (6.2), *p* < 0.001). Regarding urinary continence, ICIQ-SF scores, amount of daily used pads, and rate of continence recovery did not vary significantly between cM1-oligo patients and cM0-patients. Preoperative ICIQ-SF score was 2.3 (SD 4.6) for cM1-oligo patients and 1.1 (2.6) for cM0-patients. Postoperatively, scores raised in both subgroups to 6.4 (5.7) for cM1-oligo patients and 6.4 (5.2) for cM0 patients. The median number of daily pad use at the time of follow-up was 1.6 (SD 2.5) for cM1-oligo patients and 1.2 (1.7) for cM0 patients. At the time of follow-up, 66% of cM1-oligo patients regained continence recovery compared to 72% of cM0 patients. Analysis of time to continence rehabilitation revealed no significant difference between cM1-oligo and cM0 patients (*p* = 0.773) (Figure 1).

### 3.3. Health-Related Quality of Life

General HRQOL assessed by GHS, symptoms as well as functioning subdomains, and financial difficulty scales are summarized in Table 3. 

Preoperatively, cM1-oligo patients expressed significantly more severe symptoms concerning pain and fatigue compared to cM0-patients (18.4 (SD 27) vs. 11.2 (20.7), *p* = 0.031; 23.6 (26.4) vs. 13.7 (16.7), *p* = 0.012). In addition, cM1-oligo patients expressed higher financial difficulties scores compared to cM0-patients (11.1 (SD 23.8) vs. 2.7 (10.7), *p* < 0.001). Regarding preoperative functioning subdomains, cM1-oligo patients showed significantly lower role functioning (83.6 (SD 26.4) vs. 91.9 (19.4), *p* = 0.004), emotional (65.8 (SD 24.3) vs. 74.2 (20.9), *p* = 0.014) and social functioning (77.2 (SD 25.1) vs. 85.5 (21.6), *p* = 0.007) compared to cM0-patients. Regarding general HRQOL, cM1-oligo patients showed significantly lower preoperative GHS scores compared to cM0-patients (63.6 (SD 20.1) vs. 71.8 (S20.7), *p* = 0.004). The rate of patients with good general HRQOL (GHS ≥ 70) did not vary significantly between both groups.

At a median follow-up of 25 months, we observed significantly worse emotional functioning for cM1-oligo compared to cM0 patients (67.4 (SD 23.8) vs. 75 (22), *p* = 0.031). In addition, cM1-oligo-patients reported to be significantly less sexually active compared cM0-patients) 22.8 (SD 24.9) vs. 37.5 (29.9), *p* = 0.001). There were no significant differences regarding the remaining symptoms and functioning subscales as well as general HRQOL based on GHS.

Analysis of net changes compared to preoperative baseline is displayed in Figure 2 and Appendix A. Briefly, we observed no significant change in GHS in cM1-oligo patients (+3.2, *p* = 0.381), but a significant decrease in cM0-patients (−5, *p* = 0.001).

In the next step, we assessed the longitudinal time course of postoperative HRQOL outcomes in our matched patient cohorts. Detailed results of the respective QLQ-C30 and PR-25 subdomains at follow-up time points of 3, 12, 24, and 36 months postoperatively are summarized in Appendix A. Briefly, we found comparable HRQOL functioning subdomains courses throughout a follow-up period of 36 months. Addressing general HRQOL based on the QLQ-C30 GHS, we found increased mean GHS scores for cM0 patients 12 months postoperatively (71.5 vs. 64.2, *p* = 0.027), without any statistically significant differences at the remaining time points (Figure 3). Regarding the PR25 add-on, we found significantly increased sexually active scores for cM0 patients 24 months postoperatively, without any statistically significant differences during the remaining postoperative time course (Appendix A). 

In multivariate analysis for the primary endpoint good general HRQOL (defined as a GHS ≥ 70 following previously published cut-off values) stratified for good erectile function (IIEF5 > 18; OR 5.722, 95% CI 1.89–17.36, *p* = 0.002) and continence recovery (OR 1.671, 95% CI 1.03–2.70, *p* = 0.036), presence of oligo-metastatic bone disease did not represent a significant predictive feature for good general HRQOL at time of follow-up (OR 0.821, 95% CI 0.44–1.53, *p* = 0.536) (Table 4).

## 4. Discussion

“First, do not harm” has been a credo for every physician’s action for many centuries now. This is crucial when it comes to experimental therapeutic strategies such as cytoreductive RP for oligo-metastatic prostate cancer. Regarding the paucity of data on postoperative HRQOL following cytoreductive RP, we aimed to address this hypothesis in a contemporary well-balanced and adequately large patient cohort comparing patients undergoing cytoreductive radical prostatectomy to a control group of patients with localized disease.

Multiple studies showed a significant impact on postoperative HRQOL following RP for localized prostate cancer. The PROTECT study randomly assigned 1643 patients with localized prostate cancer to either monitoring, RP, or radiotherapy [19]. Donovan et al. evaluated HRQOL and functional outcomes with EORTC-QLQ-C30 and ICIQ questionnaires among others. The authors showed that RP had the greatest negative effect on urinary continence compared to monitoring and radiotherapy after a median follow-up of 24 months [20]. The higher mean ICIQ scores in the current study might derive from a significantly lower tumor burden in the PROTECT study group (median PSA 4.7–4.9 ng/mL, Gleason 6 in 76–78% throughout the treatment arms) and indicate a more pronounced impact on functional outcomes by higher tumor burden. In the PROTECT study, GHS after 5-years of follow-up did not vary significantly between the three treatment arms. GHS scores in the current study are lower after 25 months of follow-up compared to PROTECT, possibly deriving from shorter follow-up and higher local tumor burden. Further studies did not observe a significant impact of the surgical technique (open retropubic vs. laparoscopic robot-assisted) on long-term HRQOL following RP [13].

With regards to more advanced disease stages, patients with newly diagnosed oligometastatic prostate cancer can be offered multiple treatment options besides conventional ADT and currently approved substances enzalutamide and apalutamide showed significantly prolonged overall survival compared to baseline ADT as standard of care in this setting [1,4,5,6].

In addition, analyses from the multi-arm STAMPEDE trial showed improved overall survival in patients with low metastatic burden treated with additional radiotherapy of the primary tumor [7]. In contrast, cytoreductive RP is rarely performed today, and evidence is still subpar and is based on retrospective analyses of large registry databases that showed improved overall survival in selected patients who underwent cytoreductive RP compared to conservative therapy regimes [21,22,23,24]. Another rationale for cytoreductive RP might be local symptom control. Up to 78% of patients with metastatic prostate cancer will suffer from local complications, such as bladder outlet obstruction, ureteric obstruction, and gross hematuria at some stage [25], and it has been shown that local treatment resulted in a reduced complication rate compared to best supportive care [24].

Furthermore, data on the impact of cytoreductive RP on HRQOL is sparse. Reichard et al. retrospectively analyzed a selected group of 14 patients and reported on pre- and postoperative functional outcomes by the urinary domain of the Expanded Prostate Cancer Index Composite (EPIC) and observed a decline after a maximum follow-up of three months [26].

In the current study, we provide a matched analysis of the impact of cytoreductive RP on HRQOL in a large contemporary cohort. To our knowledge, this resembles the largest evaluation of HRQOL after cytoreductive RP to date. Hereby, we did not find significant differences of postoperative HRQOL in patients with oligometastatic disease compared to a matched control cohort with localized disease. 

In the (oligo)metastatic setting, Agarwal et al. assessed the impact of the treatment with apalutamide and ADT compared to ADT and placebo on HRQOL using the FACT-P questionnaire. The study cohort mainly comprised high-volume patients (up to 66%) and only 18% of patients in the apalutamide arm had metachronous metastatic disease with prior local therapy. The authors showed that HRQOL did not vary between both groups regarding time-to-deterioration or net baseline changes [27]. In the ARCHES study, Stenzl et al. evaluated the effect of enzalutamide combination therapy on HRQOL in metastatic hormone-sensitive prostate cancer patients [28]. The treatment arm contained 38.3% (220/574) patients classified as low-volume disease, and 25.3% had previous local therapy. HRQOL was assessed using the EORTC QLQ-PR25 and FACT-P questionnaire. Interestingly, the authors observed increased FACT-P physical well-being scores for the control arm in a subgroup of patients with low-volume disease. However, according to predefined threshold measures, this was not considered clinically meaningful. Overall, the authors found no significant differences in time-to-deterioration of HRQOL or pain between both arms as assessed by EORTC QLQ-PR25 and FACT-P scores. Although the comparability of this conservative treatment approach and our study is limited, these data are consistent with our findings. Regarding net changes from baseline, both arms of the ARCHES study showed a decline in general HRQOL over the course of therapy. The ratio of patients with low-volume disease with worsening lack of energy from baseline to end of follow-up was 48.1% for the treatment arm and 41.3% for the placebo arm. Conversely, we observed no significant change in GHS in cM1-oligo patients (+3.2, *p* = 0.381), but a significant decrease in cM0-patients (−5, *p* = 0.001). Regarding functional outcomes, Stenzl et al. found a longer time-to-deterioration in sexual activity in the placebo arm compared to the treatment arm. Likewise, we observed significantly lower total IIEF-5 scores in cM1-oligo patients compared to cM0-patients preoperatively as well as at the time of follow-up. Furthermore, cM1-oligo patients in the current study showed significantly lower postoperative QLQ-PR25 sexual activity scores compared to cM0-patient.

The current study has several limitations. Due to the retrospective design, confound patient characteristics such as specific co-morbidities were not available in detail. In addition, different staging modalities, including bone scans as well as PSMA-PET/CT was used, which might lead to a potential bias when defining oligometastatic disease stages. Furthermore, distinct information on mpMRI findings as well as molecular biomarker test results has not been assessed in the current population even though it has been shown that both modalities can work hand in hand in diagnosis in advanced prostate cancer [29]. Apart from the retrospective design, we used the non-prostate-specific EORTC-QLQ-C30 questionnaire to address HRQOL of prostate cancer patients. However, the EORTC-QLQ-C30 questionnaire is frequently used among other tumor entities and surgeries and, therefore, provides robust and comparable data. To address the specific domains of interest, sexual function and urinary continence, we complemented our evaluation by the validated ICIQ-SF and IIEF-5 questionnaires as well as the prostate-specific PR25 add-on to the QLQ-C30 questionnaire. Another potential limitation of the current study is the relatively short median follow-up of 25 months, as other studies showed significant changes in terms of sexual function and urinary continence until several years after surgery [20]. Even though this study represents the largest evaluation of HRQOL after cytoreductive RP to our knowledge, the sub-cohort of cM1-oligo patients is still relatively small compared to cM0 patients, leading to a possible risk of underpowering. Due to limitations in sample size, not all tumor burden indicators such as Gleason grade and preoperative PSA value could be included in the matching process. However, these factors did not show to be a significant predictor of good general HRQOL after RP in univariate and multivariable analysis. Finally, it has to be addressed that not all patients underwent initial PSMA-PET/CT scans prior to RP, leading to potential understaging. However, it has to be emphasized that this limitation is also inherent to currently available data from phase 3 trials in this setting. For instance, the SWOG 1802 trial (NCT 03678025) is currently recruiting [30]. Hereby, patients with metastatic prostate cancer based on conventional or PSMA-PET/CT based imaging are randomized into local therapy (RP or radiotherapy of the prostate) arm or a control arm while both arms undergo standard of care systemic androgen deprivation therapies. However, the first results are expected in 2028, and evidence from real-life data analyses will be relevant until mature data from randomized trials is publicly available.

## 5. Conclusions

In the current study, we assess HRQOL outcomes after cytoreductive RP from a comprehensive and contemporary matched patient cohort during a follow-up period of up to 36 months between patients with or without oligometastatic prostate cancer. Hereby, we found no significant differences in general HRQOL and functional outcomes between both subgroups in univariate as well as multivariable analysis indicating that cytoreductive RP can be offered safely to patients without subpar HRQOL outcomes compared to patients undergoing RP for localized disease.

## Figures and Tables

**Figure 1 cancers-13-05636-f001:**
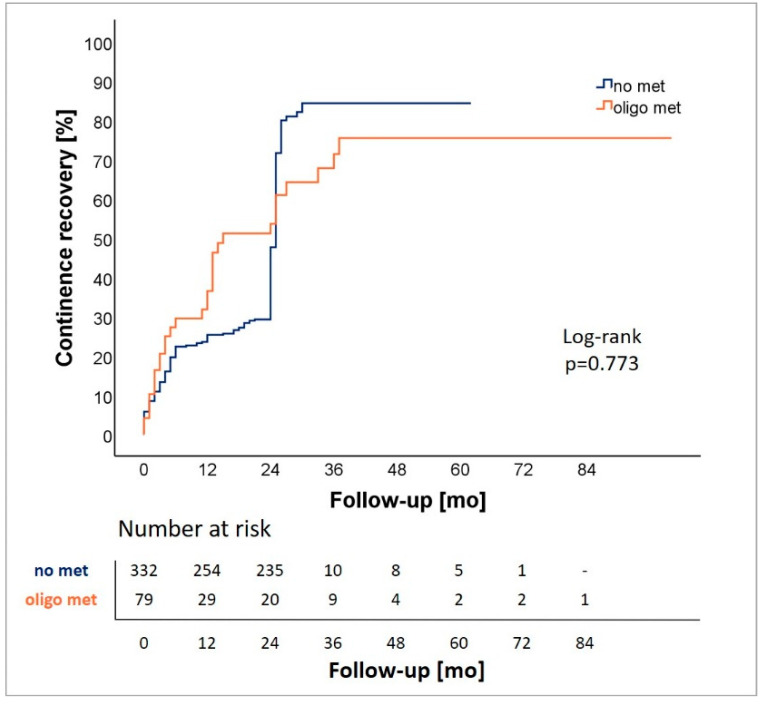
Analysis of continence recovery for patients with (oligo met) and without oligometastatic disease (no met).

**Figure 2 cancers-13-05636-f002:**
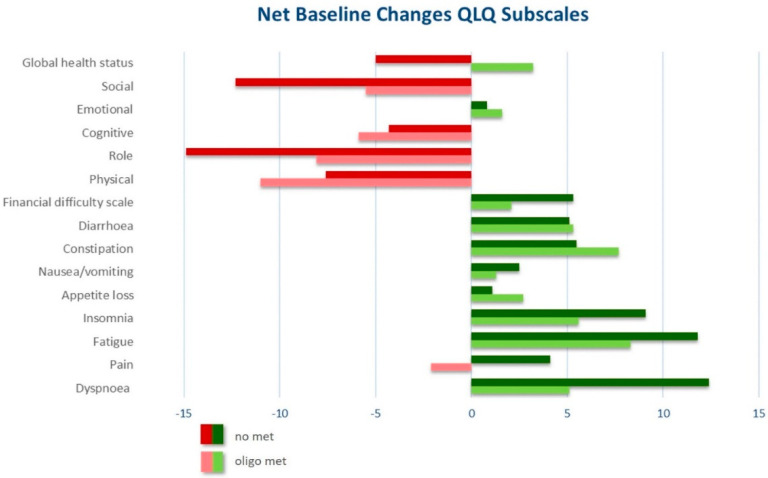
Trend illustration of net baseline changes between pre- and postoperative HRQOL.

**Figure 3 cancers-13-05636-f003:**
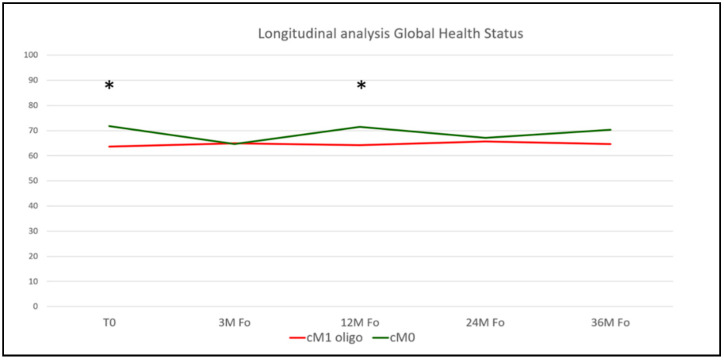
Longitudinal analysis of postoperative patient-reported Global Health Status, assessed by the validated EORTC QLQ-C30 questionnaire. Asterisks indicate *p*-values < 0.05 and were considered statistically significant. EORTC-QLQ-C30: European Organization for Research and Treatment of Cancer quality of life questionnaire-Cancer 30.

**Table 1 cancers-13-05636-t001:** Perioperative patient characteristics, Bold values indicate *p*-values < 0.05 and were considered statistically significant. Continuous values are presented as median and inter-quartile-range (IQR); categorical values are given as number (*n*; %) *BMI*: Body-Mass-Index; PSA preop: preoperative Prostate-specific Antigen level; LN: Lymphnode.

	Unmatched Cohort			Matched Cohort		
	cM1-Oligo	cM0	*p*	cM1-Oligo	cM0	*p*
No. of patients	84	1184		79	332	
Age, years (median, IQR) ^#^	66 (59, 73)	69 (66, 77)	0.039	66 (60, 72)	68 (60, 73)	0.672
BMI kg/m^2^ (median, IQR) ^#^	25.7 (24.0, 27.8)	26.1 (24.3, 28.7)	0.154	25.7 (24.0, 28)	26.2 (24.3, 29.1)	0.158
PSA preop. ng/mL (median, IQR)	25.2 (6.5, 63.3)	8.3 (5.9, 13.6)	<0.001	21.1 (6.4, 51.0)	10.9 (6.8, 20.2)	0.024
Prostate volume mL (median, IQR)	47 (40, 58)	52 (41, 66)	0.007	47.5 (39.8, 58.0)	53.5 (43.0, 67.0)	0.002
Gleason score (*n* (%))						
6	0 (0.0)	137 (11.5)	<0.001	0 (11.1)	15 (4.5)	<0.001
7a	3 (3.6)	445 (37.6)		3 (3.8)	63 (19.0)	
7b	10 (11.9)	270 (22.8)		9 (11.4)	82 (24.7)	
8	18 (21.4)	139 (11.8)		17 (21.5)	54 (16.3)	
9	46 (54.8)	180 (15.2)		42 (53.2)	105 (31.6)	
10	7 (8.3)	13 (1.1)		8 (10.2)	13 (3.9)	
pT stage (*n* (%)) ^#^						
pT2a	0 (0.0)	62 (5.3)	0.001	0 (0.0)	0 (0.0)	0.221
pT2b	0 (0.0)	24 (2.0)		0 (0.0)	0 (0.0)	
pT2c	10 (11.9)	616 (52.1)		10 (12.7)	60 (18.1)	
pT3a	15 (17.9)	265 (22.4)		15 (19.0)	88 (26.5)	
pT3b	54 (64.3)	211 (17.8)		52 (65.8)	178 (53.6)	
pT4	5 (6.0)	5 (0.4)		2 (2.5)	6 (1.8)	
Nerve sparing (*n* (%))	14 (16.7)	859 (72.6)	<0.001	13 (16.5)	183 (55.1)	<0.001
Robot-assisted RP (*n* (%))	10 (11.9)	458 (38.7)	<0.001	10 (12.7)	116 (34.9)	<0.001
Positive surgical margin (*n* (%))	64 (76.2)	322 (27.2)	<0.001	59 (74.7)	165 (49.7)	<0.001
Lymph node involvement (*n* (%))	46 (54.8)	343 (29.0)	<0.001	41 (51.9)	112 (33.7)	0.003
LN removed (median, IQR)				10 (6, 13)	11 (6, 18)	0.124
positive LN (median, IQR)				1 (0, 3)	1 (0, 1)	0.059
PSA postop. ng/mL (median, IQR)				9.5 (2.0, 70.3)	0.0 (0.0, 0.0)	<0.001

^#^ matched variables.

**Table 2 cancers-13-05636-t002:** Pre- and postoperative functional outcomes after a median follow-up of 25 months. Bold values indicate *p*-values < 0.05 and were considered statistically significant. Continuous values are presented as median and standard deviation (SD). IIEF-5: International Index of Erectile Function; ICIQ-SF: International Consultation on Incontinence Questionnaire -Short Form.

	T0			Follow-Up		
	cM1-Oligo	cM0	*p*	cM1-Oligo	cM0	*p*
Erectile function						
IIEF-5 score [mean, SD]	8.5 (10.2)	11.3 (9.9)	0.022	1.3 (4.2)	3.5 (6.2)	<0.001
IIEF-5 score 18 or more [%]	26.8	37.2	0.135	2.0	6.8	0.196
Urinary continence						
ICIQ-SF score [mean, SD]	2.3 (4.6)	1.1 (2.6)	0.081	6.4 (5.7)	6.4 (5.2)	0.970
Daily pad usage [mean, SD]	n.a.	n.a.	n.a.	1.6 (2.5)	1.2 (1.7)	0.195
Continence recovery [%]	n.a.	n.a.	n.a.	66.0	72.0	0.383

**Table 3 cancers-13-05636-t003:** Pre- and postoperative patient-reported HQOL, assessed by the validated EORTC QLQ-C30 questionnaire. Bold values indicate *p*-values < 0.05 and were considered statistically significant. Continuous values are presented as median and standard deviation (SD). *HRQOL*: health-related quality of life; EORTC-QLQ-C30: European Organization for Research and Treatment of Cancer quality of life questionnaire-Cancer 30; QLQ-PR25: European Organization for Research and Treatment of Cancer quality of life questionnaire-Prostate 25.

			Mean (SD) EORTC QLQ C30 Score					
			T0			Follow-Up		
QLQ—C 30			**cM1-Oligo**	**cM0**	** *p* **	**cM1-Oligo**	**cM0**	** *p* **
	Symptom scale							
		Dyspnoea	7.1 (16.3)	7.7 (18.7)	0.959	12.2 (21.2)	20.1 (26.6)	0.047
		Pain	18.4 (27.0)	11.2 (20.7)	0.031	16.3 (24.9)	15.3 (24.6)	0.503
		Fatigue	23.6 (26.4)	13.7 (16.7)	0.012	31.9 (25.1)	25.5 (25.3)	0.052
		Insomnia	23.0 (33.9)	18.7 (26.1)	0.706	28.6 (37.3)	27.8 (30.4)	0.590
		Appetite loss	4.8 (13.4)	3.3 (12.7)	0.250	7.5 (18.3)	4.4 (13.3)	0.272
		Nausea/vomiting	0.1 (2.2)	1.0 (4.2)	0.069	1.4 (5.7)	3.5 (10.3)	0.154
		Constipation	7.9 (23.1)	5.7 (16.4)	0.949	15.6 (27.3)	11.2 (22.8)	0.259
		Diarrhoea	7.6 (18.9)	6.9 (15.7)	0.752	12.9 (25.3)	12.0 (21.4)	0.879
	Financial difficulty scale		11.1 (23.8)	2.7 (10.7)	<0.001	13.2 (26.4)	8.0 (18.8)	0.207
	Functioning scale							
		Physical	90.4 (15.6)	93.7 (12.7)	0.130	79.4 (23.3)	86.1 (16.8)	0.177
		Role	83.6 (26.4)	91.9 (19.4)	0.004	75.5 (29.9)	77.0 (26.4.)	0.984
		Cognitive	86.8 (18.3)	89.6 (16.6)	0.272	80.9 (25.3)	85.3 (19.4)	0.398
		Emotional	65.8 (24.3)	74.2 (20.9)	0.014	67.4 (23.8)	75.0 (22.0)	0.031
		Social	77.2 (25.1)	85.5 (21.6)	0.007	71.7 (29.1)	73.2 (28.6)	0.702
	Global health status		63.6 (20.1)	71.8 (20.7)	0.004	66.8 (20.8)	66.8 (21.5)	0.959
	Global health status ≥ 70 [%]		45.6	55.2	0.186	44.0	56.0	0.811
QLQ—PR25	Urinary symptoms					27.4 (17.5)	29.2 (18.9)	0.707
		Incontinence aid				29.5 (28.8)	39.5 (34.1)	0.185
		Bowel symptoms				7.6 (12.9)	8.9 (13.4)	0.522
	Treatment symptoms					24.7 (15.9	20.7 (16.9)	0.058
	Sexually active					22.8 (24.9)	37.5 (29.9)	0.001
		Sexual functioning				55.4 (23.0)	50.8(17.6)	0.743

**Table 4 cancers-13-05636-t004:** Multivariable analysis assessing the primary endpoint good general HRQOL (defined as GHS of ≥70). Bold values indicate *p*-values < 0.05 and were considered statistically significant. Net change to preoperative status is presented, HRQOL: health-related quality of life; IIEF-5: International Index of Erectile Function; *RP*: Radical prostatectomy GHS: Global-Health status according to the European Organization for Research and Treatment of Cancer quality of life questionnaire-Cancer 30.

Multivariate Logistic Regression for Good HRQOL					
Predictive Feature for Good HRQOL	Regression Coefficient	Odds Ratio	95% CI		*p* Value
			Lower	Upper	
cM1-oligo (yes vs. no)	−0.197	0.821	0.44	1.53	0.536
IIEF-5 18 or more (yes vs. no)	1.744	5.722	1.89	17.36	0.002
Continence recovery (yes vs. no)	0.514	1.671	1.03	2.70	0.036
Nerve-sparing (yes vs. no)	0.384	1.468	0.93	2.32	0.101
Robot-assisted RP (yes vs. no)	−0.448	0.639	0.40	1.03	0.067

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
