# Peer review of "Health-Related Quality of Life following Cytoreductive Radical Prostatectomy in Patients with De-Novo Oligometastatic Prostate Cancer"

_cancers, 2021, doi:10.3390/cancers13225636_

Round 1

Reviewer 1 Report

Authors should be congratulated for the intriguing topic. Prostate cancer (PC) represents the urological milestone. All the future perspective must search novel therapeutic approaches that preserve functional outcome (improving survival outcome and quality of life of PC patients). Manuscript is well written, tables are clear and meticulous, figure are well represented. Despite the retrospective nature of the study, the methodology was robust, and the paper gave a contribution to recent surgical behaviors that was worthy of a publication. Authors satisfied the reviewers' requests, making the paper more solid.

Reviewer 2 Report

It was a pleasure reviewing the revised version of manuscript "Health-related quality of life following cytoreductive radical prostatectomy in patients with de-novo oligometastatic prostate cancer"

The authors have answered to all my quires and made satisfactory changes. I do not have any other suggestions.

Reviewer 3 Report

No additional comments to the authors from this reviewer.

This manuscript is a resubmission of an earlier submission. The following is a list of the peer review reports and author responses from that submission.

Round 1

Reviewer 1 Report

Authors should be congratulated for the intriguing topic. Prostate cancer (PCa) represents the urological milestone. All the future perspectives must search novel therapeutic approaches that preserve functional outcomes (improving survival outcome and quality of life of PCa patients). The manuscript is well written, tables are clear and meticulous, figures are well represented. Despite the retrospective nature of the study, the methodology was robust, and the paper gave a contribution to recent surgical behaviors that was worthy of publication. However, several points must be reviewed:

  1. Authors should enlighten baseline characteristics (such as comorbidity, previous TURP, biopsy naive) of the population studied. Was post-operative PSA available?
  2. How “oligometastatic PCa” diagnosis was performed? Which imaging technique was used? Was the same analysis recorded for every patient to avoid the inter-variability?
  3. How many lymph nodes were dissected when lymphadenectomy was performed? Are these data available?
  4. Are data available on PIRADS or PHI index of these PCa patients? Newly, an intriguing paper on the predictive role of PHI index for clinically significant prostate cancer (csPCa) was published (PMID: 34572950 DOI: 3390/cancers13184723). The manuscript is worthy of being read. We must consider the future horizons: firstly, new stratification of PCa patients (using new markers or imaging-integrated approaches) is necessary to create patients tailored-therapies, understanding the potential aggressiveness of disease and quality of life after treatment.

Author Response

Authors should be congratulated for the intriguing topic. Prostate cancer (PCa) represents the urological milestone. All the future perspectives must search novel therapeutic approaches that preserve functional outcomes (improving survival outcome and quality of life of PCa patients). The manuscript is well written, tables are clear and meticulous, figures are well represented. Despite the retrospective nature of the study, the methodology was robust, and the paper gave a contribution to recent surgical behaviors that was worthy of publication. However, several points must be reviewed:

We thank the reviewer for his encouraging comments.

1.Authors should enlighten baseline characteristics (such as comorbidity, previous TURP, biopsy naive) of the population studied. Was post-operative PSA available?

According to the reviewer’s valuable comments, we have included more parameters in the baseline characteristics analysis of our matched patient cohort. We summarized the data in a novel supplementary table and included a history of TUR-P, history of hernia surgery as well as ASA score as a potential surrogate for comorbidities. Due to the retrospective nature of the analysis, we were not able to include specific co-morbidities. We address this shortcoming in the revised limitations section of our manuscript.

2.How “oligometastatic PCa” diagnosis was performed? Which imaging technique was used? Was the same analysis recorded for every patient to avoid the inter-variability?

In order to address this comment, we have reviewed our matched patient cohorts and have included the information of staging modalities in the current version of the manuscript. These analyses revealed that conventional imaging as well as PSMA-PET/CT was used for staging purposes. We address this limitation in the revised manuscript. However, since evidence of oligometastatic prostate cancer is mainly based on conventional imaging, we believe that our data represents current patient collectives.

3.How many lymph nodes were dissected when lymphadenectomy was performed? Are these data available?

Based on the reviewer’s comments, we have included the median (IQR) number of removed as well as positive lymph nodes in the revised table 1.

4.Are data available on PIRADS or PHI index of these PCa patients? Newly, an intriguing paper on the predictive role of PHI index for clinically significant prostate cancer (csPCa) was published (PMID: 34572950 DOI: 3390/cancers13184723). The manuscript is worthy of being read. We must consider the future horizons: firstly, new stratification of PCa patients (using new markers or imaging-integrated approaches) is necessary to create patients tailored-therapies, understanding the potential aggressiveness of disease and quality of life after treatment.

Thank you very much for the comment and the respective reference. Unfortunately, PIRADS scores as well as molecular biomarker information is not available for our patient collective. We address this limitation in the revised manuscript and included the suggested reference in the discussion.

Reviewer 2 Report

It was a pleasure reviewing the manuscript "Health-related quality of life following cytoreductive radical prostatectomy in patients with de-novo oligometastatic prostate cancer"

This manuscript raises several questions:-

  1. Why were patients with de-novo disease selected for cytoreductive radical prostatectomy? Were they enrolled on trial or why these off-label treatment was chosen for these select patients?
  2. By definition-- Propensity score matching attempts to reduce the bias due to confounding variables that could be found in an estimate of the treatment effect obtained from simply comparing outcomes among units that received the treatment versus those that did not. By just using three variables age, BMI and pT stage the authors are not doing propensity matching but just matching on 3 variables. They need to include all variables which a surgeon will take into account before doing radical prostatectomy. It will include, Gleason, PSA, co-morbidities like HTN, heart failure, diabetes, perioperative risk factors, metastatic burden, year of treatment, even insurance and social variables etc. As seen in Table 1 many of these variables are not matched and this analysis is not correct.
  3. How did the authors account for differences in time duration for these procedures and HRQOL?

Author Response

It was a pleasure reviewing the manuscript "Health-related quality of life following cytoreductive radical prostatectomy in patients with de-novo oligometastatic prostate cancer"

Thank you very much for your kind comments!

This manuscript raises several questions:-

  1. Why were patients with de-novo disease selected for cytoreductive radical prostatectomy? Were they enrolled on trial or why these off-label treatment was chosen for these select patients?

Thank you very much for this comment. We performed cytoreductive RP as an individual therapy concept in a multimodal setting for patients willing to undergo surgery for oligometastatic disease. Our center participated in the G-RAMPP (NCT 02454543) trial investigating this topic, however this trial has been stopped prematurely. All patients were fully aware of the experimental nature of the procedure and were informed about alternative evidence-based regimen. It has to be emphasized that the vast majority of the analyzed patients underwent RP before publication of the results of STAMPEDE and HORRAD trial. We have specified this information in the revised version of the manuscript.

  1. By definition-- Propensity score matching attempts to reduce the bias due to confounding variables that could be found in an estimate of the treatment effect obtained from simply comparing outcomes among units that received the treatment versus those that did not. By just using three variables age, BMI and pT stage the authors are not doing propensity matching but just matching on 3 variables. They need to include all variables which a surgeon will take into account before doing radical prostatectomy. It will include, Gleason, PSA, co-morbidities like HTN, heart failure, diabetes, perioperative risk factors, metastatic burden, year of treatment, even insurance and social variables etc. As seen in Table 1 many of these variables are not matched and this analysis is not correct.

Thank you very much for this important comment. We agree that a more holistic matching process would be highly desirable. Due to the limited sample size of our oligo-met cohort, we limited our matching analysis on three parameters that have already been shown to affect postoperative HRQOL. Unfortunately, some of the parameters requested by the reviewer cannot be satisfactory obtained from our database and medical charts. We have addressed this issue in the revised limitations section and have renamed “propensity score matching” with “matching” throughout the manuscript. In addition, we have specified the rationale behind the matching process in our revised M&M section. We hope the reviewer agrees with these explanations.

  1. How did the authors account for differences in time duration for these procedures and HRQOL?

Based on this very valuable comment, we have completely revised the HRQOL section and included a novel analysis considering the respective time points of HRQOL follow-up retrieval. We have summarized the data in a newly-generated supplementary table that includes all respective subscales and created a novel figure that highlights the longitudinal analysis of the global health status which might act as a surrogate for general HRQOL.

Reviewer 3 Report

This is an interesting retrospective study on the potential role of RP in pts with oligometastatic PC vs non-metastatic PC. Major issues that need to be addressed include:

  • the SWOC 1802 study is a randomized phase 3 addressing the same question in terms of overall outcomes and is currently ongoing. This should be discussed.
  • there are important and relevant clinical information missing, including systemic therapies (ADT, AR signaling inhibitors, chemotherapy), local pelvic/prostate irradiation, sites of mets (nodal vs bone vs visceral) comorbidities in the patient population. These are all important and must be disclosed as they can also significantly affect QoL parameters/metrics.

Author Response

This is an interesting retrospective study on the potential role of RP in pts with oligometastatic PC vs non-metastatic PC. Major issues that need to be addressed include:

Thank you very much for your kind comment!

the SWOC 1802 study is a randomized phase 3 addressing the same question in terms of overall outcomes and is currently ongoing. This should be discussed.

Thank you very much for this remark. We have included the study in the discussion section of the revised manuscript.

there are important and relevant clinical information missing, including systemic therapies (ADT, AR signaling inhibitors, chemotherapy), local pelvic/prostate irradiation, sites of mets (nodal vs bone vs visceral) comorbidities in the patient population. These are all important and must be disclosed as they can also significantly affect QoL parameters/metrics.

Based on your very valuable comment, we have highlighted the summary of adjuvant and salvage therapies in the revised version of our manuscript. Since we were not able to retrieve co-morbidities in a reliable manner, we have also included the ASA score as well as history of TURP and hernia surgery in the revised version of the manuscript. Based on your comments, we have also specified the inclusion criteria of our study cohort. We have focused on patients with clinical suspicion of bone (oligo) metastases. Patients with nodal-only or visceral metastases were excluded from further analyses.